# Tailored Teaching with Balanced Difficulty: Elevating Reasoning in Multimodal Chain-of-Thought via Prompt Curriculum

## Abstract

The effectiveness of Multimodal Chain-of-Thought (MCoT) prompting is often limited by the use of randomly or manually selected examples. These examples fail to account for both model-specific knowledge distributions and the intrinsic complexity of the tasks, resulting in suboptimal and unstable model performance. To address this, we propose a novel framework inspired by the pedagogical principle of "tailored teaching with balanced difficulty". We reframe prompt selection as a prompt curriculum design problem: constructing a well ordered set of training examples that align with the model's current capabilities. Our approach integrates two complementary signals: (1) model-perceived difficulty, quantified through prediction disagreement in an active learning setup, capturing what the model itself finds challenging; and (2) intrinsic sample complexity, which measures the inherent difficulty of each question–image pair independently of any model. By jointly analyzing these signals, we develop a difficulty-balanced sampling strategy that ensures the selected prompt examples are diverse across both dimensions. Extensive experiments conducted on five challenging benchmarks and multiple popular Multimodal Large Language Models (MLLMs) demonstrate that our method yields substantial and consistent improvements and greatly reduces performance discrepancies caused by random sampling, providing a principled and robust approach for enhancing multimodal reasoning.

## 1 Introduction

Multimodal Large Language Models (MLLMs) Yin et al. (2024); Liu et al. (2023a) have emerged prominent research focus alongside the rapid advancement of artificial intelligence. Built upon powerful foundation language models Touvron et al. (2023); Bai et al. (2023), MLLMs leverage cross-modal alignment mechanisms to achieve understanding and processing of information across multiple modalities, including text, images, videos, and audio. A typical approach to deploying MLLMs is the in-context learning paradigm Brown et al. (2020); Xie & Min (2022), which drives models to perform predictions by providing a large number of instructions and input-output pair examples. Chain-of-Thought Wei et al. (2022); Zhou et al. (2022) enhances the logical reasoning capability of models by constructing examples that decompose complex problems into step-by-step subproblems and solve them sequentially. Multimodal Chain-of-Thought (MCoT) Zhang et al. (2023), on the other hand, further extends this core idea to the application scenarios of multimodal large language models, achieving an improvement in cross-modal reasoning capabilities.

MCoT encourages large language models to perform multi-step reasoning by providing explicit problem-solving rationales, rather than mapping questions directly to answers. This approach has been effectively applied to scientific question answering across domains such as natural sciences, linguistic sciences, and social sciences. However, as illustrated in Figure 1(a), MCoT prompt examples are typically randomly selected or manually crafted without considering the model's internal knowledge distribution or the characteristics of the dataset. This leads to prompts that are unstable and insufficiently tailored to the model or task. As a result, MCoT prompting often suffers from poor generalization, hallucinated outputs, and inconsistent performance. To enhance the specificity and effectiveness of prompt examples, Auto-CoT Zhang et al. (2022) constructs prompts by selecting representative examples via clustering, as shown in Figure 1(b). However, due to the inherent

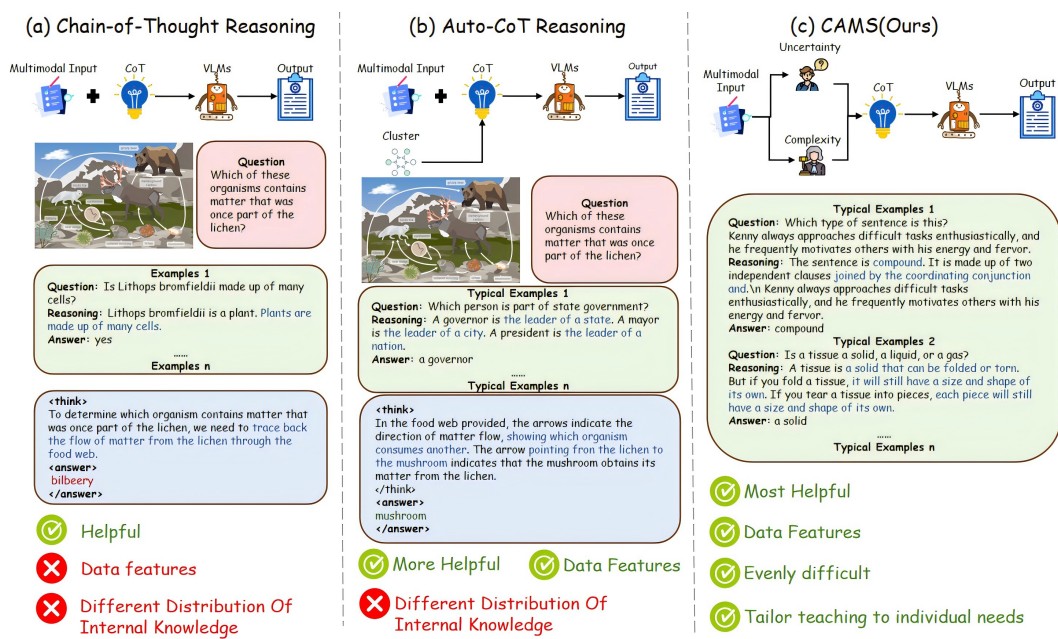

Figure 1: Illustration of the motivation and key highlights of our proposed framework. (a) CoT uses random/manual prompts without analyzing model knowledge distribution or dataset features; (b) Auto-CoT uses clustering for representative prompts but ignores inter-model knowledge differences; (c) CAMS (Ours) screens optimal prompts via active learning uncertainty and complexity analysis, balancing difficulty to enhance effectiveness. We adopt the same multimodal input consisting of images and questions as in (a) and (b) for CAMS.

differences between modalities, the effectiveness of Auto-CoT in multimodal scenarios is limited. It fails to adequately account for the distributional differences in internal knowledge across different models.

These challenges raise a critical question: how to selectively identify the most effective multimodal prompt examples? In human learning, individuals often compile personalized sets of challenging problems ranging from basic misunderstandings due to knowledge gaps to complex tasks that require multi-step reasoning. Inspired by the idiom **"Tailor teaching to individual needs"**, we treat each multimodal large model as a unique learner and select prompt examples through two dimensions: uncertainty analysis and complexity evaluation.

We propose CAMS (Complexity-Guided Active Multimodal CoT Sampling), a novel joint selection framework that integrates active learning and data complexity assessment. Rather than relying on randomly selected or manually crafted MCoT examples, CAMS constructs a tailored "error sets" for each model based on the training sets of various reasoning tasks, aiming to enhance the models' performance on test sets. As shown in Figure 1(c), CAMS identifies highly targeted, effective, and difficulty-balanced prompt examples by jointly analyzing sample uncertainty and complexity. We conduct experiments on five benchmark datasets and three multimodal large models. The results show that CAMS both improves model performance and greatly reduces accuracy variability caused by random prompt selection. Our key contributions are as follows:

- We introduce CAMS, the first framework to select prompt examples based on both model-internal knowledge and dataset characteristics.

- We demonstrate that effective prompting requires a balance of easy and hard examples; neither extreme is sufficient on its own.

- CAMS greatly reduces the instability of traditional prompt selection methods and enables MLLMs to achieve stable, high performance on complex reasoning tasks.

## 2 RELATED WORK

### 2.1 PROMPT SELECTION BASED ON ACTIVE LEARNING

Active learning is a machine learning paradigm focused on maximizing model performance using the fewest possible labeled samples. It aims to identify the most informative unlabeled data points for annotation, thereby reducing labeling costs while maintaining high accuracy. Active learning methods are typically grouped into the following three categories based on how unlabeled data is queried: membership query synthesis Angluin (1988); King et al. (2004), where the model generates new instances for labeling; stream-based selective sampling Dagan & Engelson (1995); Krishnamurthy (2002), where data points are evaluated one at a time for potential labeling; and pool-based sampling Lewis (1995), where the model selects the most informative samples from a large pool of unlabeled data.

Active Prompt Diao et al. (2023) applies active learning principles to prompt selection by quantifying uncertainty through prediction disagreement metrics (e.g., variance, entropy, disagreement) and choosing high-uncertainty samples as prompt examples. However, Active Prompt focuses solely on model's prediction disagreement with samples (i.e., distributional differences in model-internal knowledge) without considering sample complexity. Our approach introduces a data complexity evaluator that assesses the inherent difficulty of samples. This allows for more customized and effective prompt selection, combining insights from both model knowledge and dataset characteristics.

### 2.2 CHAIN-OF-THOUGHT IN VISUAL QUESTION ANSWERING

The Multimodal Chain of Thought (CoT) technique is widely adopted to enhance the multi-step reasoning abilities of large language models (LLMs). Its core idea is to guide models to generate intermediate reasoning steps that help them solve complex problems more effectively. Benchmarks such as VQA Antol et al. (2015), VQAv2 Goyal et al. (2017), OK-VQA Marino et al. (2019), A-OKVQA Schwenk et al. (2022), and ScienceQA Lu et al. (2022) provide structured visual question answering (VQA) tasks across various domains, including natural science, social science, semantics, and everyday reasoning. For complex reasoning tasks, recent approaches like MCoT Zhang et al. (2023), Auto-CoT Zhang et al. (2022), Self-Consistency Wang et al. (2022), and Active Prompt Diao et al. (2023) leverage carefully designed prompt examples to improve model performance.

Despite these advancements, many methods Wei et al. (2022); Wang et al. (2022); Zhou et al. (2022) rely on either randomly selected or manually crafted prompt examples. These examples often fail to align with the specific demands of individual VQA tasks, limiting model performance. In particular, they tend to overlook key factors such as the distributional characteristics of the model's internal knowledge and the multimodal nature of VQA tasks, focusing primarily on unimodal scenarios. To address these limitations, our framework jointly considers the model's uncertainty about the dataset and the intrinsic complexity of each example. By integrating both model-centric and data centric perspectives, we dynamically construct prompt examples that are better tailored to the model's current capabilities and the reasoning requirements of the task, leading to more effective and adaptive prompting in visual question answering.

## 3 METHODOLOGY

Figure 2 illustrates the three core modules of CAMS: **(i) Characterization of Multimodal Model Internal Knowledge**, which quantifies the model's predictive uncertainty through multiple independent samplings of the model, revealing the distribution characteristics of the model's internal knowledge; **(ii) Complexity-Based Dataset Feature Estimation**, which is used to quantitatively evaluate dataset characteristics; **(iii) Examples Sampling Strategy**, which incorporates model uncertainty indicators, dataset complexity scores, and the "easy-hard" selection principle to dynamically identify the most representative and effective prompt examples for the target model.

### 3.1 PROBLEM DEFINITION

We define a visual question answering dataset as $D = \{(x_i, v_i, y_i)\}_{i=1}^{N}$, where $x_i$ represents the linguistic text query, $v_i$ represents the corresponding visual image input, $y_i$ denotes the ground truth,

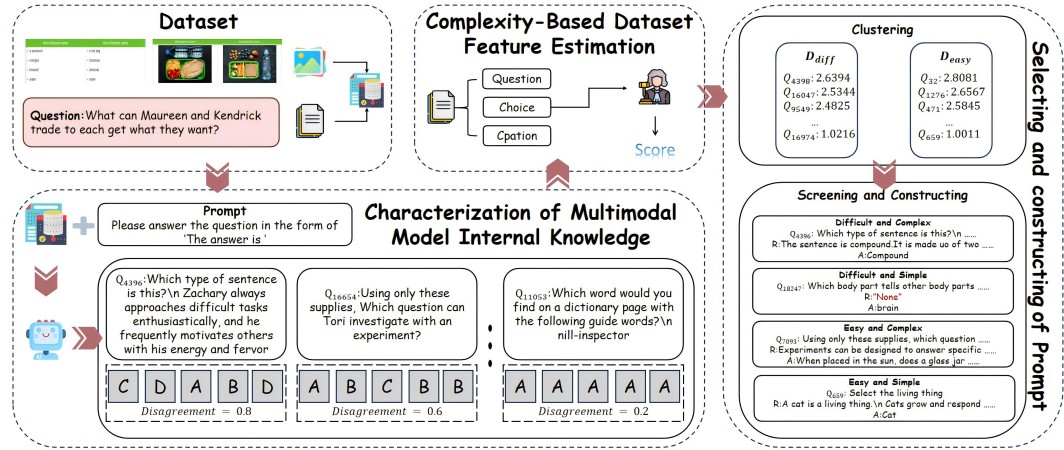

Figure 2: The illustration of our CAMS framework. **Dataset** consists of multimodal inputs of images and text. **Complexity-Based Dataset Feature Estimation** calculates complexity by integrating question text and image captions to evaluate dataset characteristics. **Analysis of Multimodal Model Internal Knowledge** reveals the distribution of the model's internal knowledge through the uncertainty of the model's multiple predictions.

and N is the total number of test samples. The goal of the prompt optimization task is to search for the optimal prompt $p^*$ that maximizes the performance $\mathcal{A}(\cdot)$ of large language models (LLMs) on a given task. This task can be formally defined as:

$$p^* = \underset{p \in \mathcal{P}_{space}}{argmax} \sum_{i=1}^{N} \mathcal{S}(\mathcal{A}(x_i, v_i; p), y_i) \tag{1}$$

where $\mathcal{P}_{space}$ denotes the set of all possible prompts (automatically selected or manually crafted), and $\mathcal{S}(\cdot)$ represents the corresponding evaluation metric.

### 3.2 CHARACTERIZATION OF MULTIMODAL MODEL INTERNAL KNOWLEDGE

#### 3.2.1 DISAGREEMENT.

We define the training sample set as $\mathcal{D}_{train} = \{q_1, q_2, ..., q_n\}$, where $q_i$ denotes an unlabeled sample and $n$ is the total number of training samples. We instruct the MLLM $\mathcal{F}_\theta(\cdot)$ to sample from the training set $\mathcal{D}_{train}$, generating the sample results:

$$\mathcal{F}_\theta^i = \{\mathcal{F}_\theta^i(q_1), \mathcal{F}_\theta^i(q_2), ..., \mathcal{F}_\theta^i(q_n)\} \tag{2}$$

where $\mathcal{F}_\theta(q_i)$ represents the model's response to sample $q_i$. After performing k sampling iterations, the final results are denoted as :

$$\mathcal{A} = \{a_1, a_2, ..., a_k\} \tag{3}$$

where $a_i = \{\mathcal{F}_\theta^1(q_i), \mathcal{F}_\theta^2(q_i), ..., \mathcal{F}_\theta^k(q_i)\}$ represents the k sampling outcomes for sample $q_i$, $\mathcal{F}_\theta^j(q_i)$ denotes the j-th sample generated by the model for input, and $\mathcal{A}$ represents the collection of $k$ sampling results for all samples. We then compute unique answer via set operations to remove duplicates, yielding $k'$ unique items $a_i = \{\mathcal{F}_\theta^1(q_i), \mathcal{F}_\theta^2(q_i), ..., \mathcal{F}_\theta^{k'}(q_i)\}$, where $\frac{k'}{k} \in (0, 1.0]$ denotes the disagreement metric $u$. A larger value of the disagreement metric $u$ indicates a higher difficulty level for the sample.

#### 3.2.2 CLUSTERING.

After assigning each training sample a corresponding uncertainty metric $u$, we perform clustering on all training samples based on the numerical values of $u$, yielding a new training sample set.

Specifically, we classify samples with $u \in (0.5, 1.0]$ as difficult questions; conversely, samples with $u \in (0, 0.5]$ are classified as easy questions.

$$\mathcal{D}_{train}^{diff} = \{\mathcal{D}_{difficulty}, \mathcal{D}_{easy}\},$$
$$\mathcal{D}_{difficulty} = \{\mathcal{D}_{u_i}\}_{0.5 < u_i \leq 1.0}, \quad (4)$$
$$\mathcal{D}_{easy} = \{\mathcal{D}_{u_j}\}_{0 < u_j \leq 0.5}.$$

where $u_i$ represents the specific disagreement value, and $\mathcal{D}_{u_i}$ represents the set of samples whose disagreement metric equals $u_i$.

### 3.3 COMPLEXITY-BASED DATASET FEATURE ESTIMATION

**Training of the Complexity Scorer.** We introduce an evolution-based metric, Evol Complexity. We define a small-scale seed dataset $\mathcal{D} = \{(I_1^{(0)}, R_1^{(0)}), (I_2^{(0)}, R_2^{(0)}), ..., (I_N^{(0)}, R_N^{(0)})\}$, where $(I_i^{(0)}, R_i^{(0)})$ denotes instruction-response pairs and $N$ denotes the number of instruction-response pairs. For each instruction sample $I$, we enhance its complexity through Technique $\mathcal{F}_\alpha(\cdot)$, which involves adding constraints, specification, and increasing reasoning steps. After $N$ iterations, a set of instructions with varying complexities is obtained:

$$\{(I_i^{(0)}, R_i^{(0)}), (I_i^{(1)}, R_i^{(1)}), ..., (I_i^{(M)}, R_i^{(M)})\} \quad (5)$$

where $I_i^{(m)} = \mathcal{F}_\alpha(I_i^{(m-1)})$, $M$ denotes the number of sample evolution iterations and is set to 5. Further, we utilize the scoring function $\mathcal{S}(\cdot)$ (i.e. ChatGPT) to rate and rank these 6 samples, generating a set of instructions with scoring labels:

$$\{(I_i^{(j)}, \mathcal{S}(I_i^{(j)}), R_i^{(j)})\}_{j=0}^M \quad (6)$$

Composed of these labeled instruction groups, a dataset is constructed to train `LLaMA` as the complexity scorer. We use the `Llama2-7B` model trained on the **"6k SFT + 10k DPO"** dataset as the final complexity scorer. Experiments demonstrate that this model can effectively classify sample complexity, with an AlpacaEval score of 90.06% Liu et al. (2023b).

**Computation of Complexity.** This scheme aims to convert images in multimodal data into text form (image captions) and integrate them with question texts to form unimodal inputs, enabling the reuse of existing unimodal complexity scorers. Specifically, we observe that each image in every multimodal dataset is accompanied by a corresponding textual description (i.e., a caption), and we thus decide to make full use of these image captions. The image captions are then integrated with the corresponding questions and options to form complete unimodal inputs in pure text. We adopt the concatenation format of "*question text + option text + image caption*", using the delimiter $'\backslash n'$ to ensure that the scorer can recognize the text representation of visual information. Finally, the integrated text data is fed into the complexity scorer, which outputs the corresponding sample complexity score.

### 3.4 STRATEGIES FOR SELECTING EXAMPLES

We employ a difficulty-balanced sample selection strategy. Specifically, for subsets of questions categorized as difficult and error-prone, we select an equal number of high-complexity and low-complexity questions; this balanced selection criterion is equally applied to subsets of simple and fundamental questions. This selection strategy fully incorporates two dimensions: uncertainty and complexity. **!!!! It is important to note that all test samples within a given dataset share the prompt examples selected through this method, rather than each test sample using a distinct set of prompt examples.**

## 4 EXPERIMENTS

In this section, we conduct extensive experiments to validate the effectiveness of CAMS (Complexity-Guided Active Multimodal CoT Sampling). Our experiments aim to address the following core questions:

**Q1.** Compared with existing baseline methods, can CAMS improve the accuracy of final answers?

**Q2.** Can CAMS eliminate the instability of randomly selecting prompt examples?

**Q3.** Are the designs of our two modules (active learning and complexity scoring) both meaningful?

**Q4.** Can CAMS enhance the accuracy of multimodal large models in subdivided domains?

**Q5.** Why should we select examples with uniform uncertainty for multimodal large models?

## 4.1 EXPERIMENTAL SETTINGS

### 4.1.1 DATASETS AND METRICS.

Following the standard evaluation settings in MLLMs reasoning research,we conduct testing on five popular benchmarks: ScienceQA Lu et al. (2022), A-OKVQA Schwenk et al. (2022), OK-VQA Marino et al. (2019), VQAv2 Goyal et al. (2017) and TextVQA Singh et al. (2019). The problem designs in these datasets include both multiple-choice and open-ended answer formats, and we follow the practice Liu et al. (2024); Li et al. (2024); Bai et al. (2023) of using accuracy as the metric.

### 4.1.2 BASELINES AND MODEL VARIANTS.

In our experiment, the following five methods are used as the main baselines: Chain-of-thought (ZS-CoT) Wei et al. (2022), Few shot CoT (FS-CoT), Self-consistency (Self-Con) Wang et al. (2022), Auto-CoT Zhang et al. (2022) and Active Prompt (Active-Pro) Diao et al. (2023). FS-CoT uses the same annotation process as our method, the only difference being that it randomly selects questions from the training data for annotation. We select multiple cutting-edge MLLMs as experimental models, including `llama3.2-vision:11b`, `llava:7b`, and `Qwen2.5-VL:7b`.

## 4.2 IMPLEMENTATION DETAILS

### 4.2.1 HYPERPARAMETERS.

In the training set sampling phase, we follow the optimal value of 10 Diao et al. (2023) for the number of sampling iterations. During both the sampling and the inference phases of the experiments, the temperature is uniformly set at $0.5$. Unless otherwise specified, the multimodal large language models used in the experiments are `llama3.2-vision:11b`, `llava:7b`, and `Qwen2.5-VL:7b`.

### 4.2.2 UNCERTAINTY ASSESSMENT.

In the experimental process, we adopt a zero-shot sampling strategy, which does not rely on additional examples or guidance information. For ScienceQA, A-OKVQA, and VQA, we perform sampling on the complete training sample sets, while for the VQAv2 and TextVQA datasets, we only sample 10,000 samples[1]. For the sampling frequency parameter, We follow the optimal value Diao et al. (2023) to set $k = 10$. We consistently use "Disagreement"[2]—a more intuitive and accurate method—as the uncertainty metric.

### 4.2.3 CONSTRUCTING EXAMPLES.

We focus on the innovating and optimizating strategies for prompt example selection. To reduce human labor as much as possible, we eliminate the need for manual annotation of prompt examples. Instead, we construct prompt instances by directly concatenating the question, reasoning, and answer from the dataset. The solution processes in the dataset are carefully crafted and rigorously screened by the authors, considering domain knowledge or task-specific styles. This ensures the generalization ability of our method to a certain extent. Details are provided in Appendix A.1.

---

[1]Because the training sets of these two datasets are too large, full sampling would require substantial computational and time costs.

[2]In preliminary experiments, variance and entropy exhibit suboptimal performance in multimodal scenarios.

Table 1: Overall results (%) on five benchmarks. In each setting, the best results are displayed in bold and italics, with gray shading indicating the degree of improvement compared to ZS-CoT.

| METHOD | Datasets | | | | | Avg. |
|---|---|---|---|---|---|---|
| | ScienceQA | A-OKVQA | OK-VQA | VQAv2 | TextVQA | |
| **Llama3.2-vision:11b** | | | | | | |
| ZS-CoT | 38.08 ↑0.000 | 47.42 ↑0.000 | 28.74 ↑0.000 | 57.45 ↑0.00 | 37.52 ↑0.000 | 41.842 ↑0.0000 |
| FS-CoT | 60.42 ↑22.34 | 55.88 ↑8.460 | 50.92 ↑22.18 | 59.95 ↑2.50 | **65.00** ↑27.48 | 58.434 ↑16.592 |
| Auto-CoT | 39.07 ↑0.990 | **59.82** ↑12.40 | 48.13 ↑19.39 | 63.57 ↑6.12 | 61.14 ↑23.62 | 54.346 ↑12.504 |
| Active-Pro | 40.44 ↑2.360 | 57.55 ↑10.13 | 46.86 ↑18.12 | 60.12 ↑2.67 | 58.86 ↑21.34 | 52.776 ↑10.924 |
| Self-Con | 50.00 ↑11.92 | 54.41 ↑6.990 | 43.56 ↑14.82 | **66.04** ↑8.59 | 54.85 ↑17.33 | 53.772 ↑11.930 |
| Ours | **68.22** ↑30.14 | 59.39 ↑11.97 | **51.89** ↑23.15 | 61.89 ↑4.44 | 62.76 ↑25.24 | **60.830** ↑18.988 |
| **Llava:7b** | | | | | | |
| ZS-CoT | 41.10 ↑0.000 | 59.39 ↑0.00 | 3.910 ↑0.000 | 7.820 ↑0.000 | 0.300 ↑0.000 | 22.504 ↑0.0000 |
| FS-CoT | 59.79 ↑18.69 | 62.31 ↑2.92 | 31.82 ↑27.91 | 29.36 ↑21.54 | 20.43 ↑20.13 | 40.742 ↑18.238 |
| Auto-CoT | 56.12 ↑15.02 | 62.18 ↑2.79 | 34.34 ↑30.43 | 30.80 ↑22.98 | 20.48 ↑20.18 | 40.784 ↑18.280 |
| Active-Pro | 57.20 ↑16.10 | 61.39 ↑2.00 | 34.10 ↑30.19 | 30.69 ↑22.87 | 20.58 ↑22.87 | 40.792 ↑18.288 |
| Self-Con | 57.91 ↑16.81 | 62.10 ↑2.71 | 14.06 ↑10.15 | 11.00 ↑3.180 | 9.140 ↑8.840 | 30.842 ↑8.3380 |
| Ours | **62.53** ↑21.43 | **63.41** ↑4.02 | **34.46** ↑30.55 | **33.07** ↑25.25 | **20.78** ↑20.48 | **42.850** ↑20.346 |
| **Qwen2.5-VL:7b** | | | | | | |
| ZS-CoT | 40.16 ↑0.000 | 39.39 ↑0.000 | 6.910 ↑0.000 | 5.180 ↑0.000 | 1.900 ↑0.00 | 18.708 ↑0.0000 |
| FS-CoT | 83.44 ↑43.28 | 71.35 ↑31.96 | 20.86 ↑13.95 | 29.98 ↑24.80 | 4.330 ↑2.43 | 41.992 ↑23.284 |
| Auto-CoT | 74.09 ↑33.93 | 69.78 ↑30.39 | 19.90 ↑12.99 | 30.49 ↑25.31 | 4.020 ↑2.12 | 39.656 ↑20.948 |
| Active-Pro | 77.79 ↑37.63 | 71.26 ↑31.87 | 21.70 ↑14.79 | 28.92 ↑23.74 | 4.040 ↑2.14 | 40.742 ↑22.034 |
| Self-Con | 74.70 ↑34.54 | 64.45 ↑25.06 | 9.540 ↑2.430 | 7.610 ↑2.430 | 2.420 ↑2.12 | 31.744 ↑13.036 |
| Ours | **84.08** ↑43.92 | **71.79** ↑32.40 | **24.52** ↑17.61 | **31.14** ↑25.96 | **4.580** ↑2.68 | **43.222** ↑24.514 |

## 4.3 MAIN RESULTS

### 4.3.1 FOR Q1: CAMS CONSISTENTLY OUTPERFORMS NEARLY ALL BASELINE METHODS.

Among the three models — `Llama3.2-vision:11b`, `Llava:7b`, and `Qwen2.5-VL:7b` — the average accuracy (Avg.) of the proposed method consistently outperforms nearly all baseline methods. As shown in Table 1, taking `Llama3.2-vision:11b` as an example: our method achieves an average score approximately 45.38 points higher than that of ZS-CoT, and more than 5 points higher than those of Auto-CoT, Active Prompt, and self-consistency. The best-performing FS-CoT is around 2 points lower than CAMS. This indicates that CAMS can stably enhance the performance of multimodal large models on complex multimodal reasoning tasks, demonstrating robust effectiveness compared to existing baseline methods across diverse model configurations.

### 4.3.2 FOR Q2: CAMS CAN GREATLY REDUCE ACCURACY INSTABILITY CAUSED BY RANDOMLY SELECTING PROMPT EXAMPLES.

To ensure the fairness and reliability of the experiments, five tests are conducted on the FS-CoT method using different random seeds on the test set, with the average accuracy across the five tests selected as the final result. Figure 3 illustrates the specific performance of FS-CoT in these five tests. The line chart reveals that the random selection strategy exhibits significant instability, with substantial fluctuations in accuracy across tests — the difference between the highest and lowest accuracy exceeds 5. CAMS not only significantly reduces the impact of randomness on performance but also consistently achieves above-average accuracy, outperforming the random selection strategy across the board.

### 4.3.3 FOR Q3: BOTH THE UNCERTAINTY ANALYSIS AND COMPLEXITY EVALUATION MODULES CAN EFFECTIVELY IMPROVE ACCURACY.

Table 2 presents the findings of the ablation experiments. To clarify the design of each method compared in the table, key definitions are as follows: ZS-CoT solely employs the simple prompt

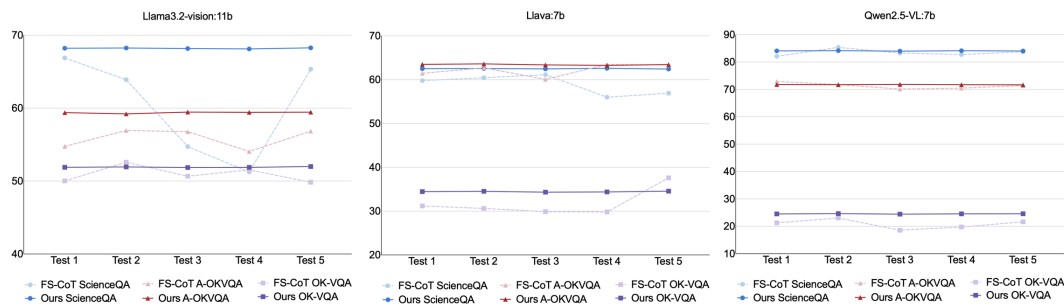

Figure 3: Accuracy fluctuations across five tests of FS-CoT and CAMS, where Test 1–5 denote the serial numbers of each test.

Table 2: Results of ablation study (%) on five benchmarks. **Unc-Eva** selects examples randomly only from different uncertainty categories. **Com-Eva** selects only an equal number of high-complexity and low-complexity samples.

| | Module | | Datasets | | | | | Avg. |
|---|---|---|---|---|---|---|---|---|
| | Unc-Eva | Com-Eva | ScienceQA | A-OKVQA | OK-VQA | VQAv2 | TextVQA | |
| | | | **Llama3.2-vision:11b** | | | | | |
| (a) | ZS-CoT | | 38.08 | 47.42 | 28.74 | 57.45 | 37.52 | 41.84 |
| | | | ↓ 48.58% | ↓ 20.15% | ↓ 44.61% | ↓ 7.17% | ↓ 40.22% | ↓ 31.22% |
| (b) | ✔ | | 63.40 | 56.68 | 43.65 | 59.75 | 61.32 | 56.96 |
| | | | ↓ 7.07% | ↓ 4.56% | ↓ 15.88% | ↓ 3.46% | ↓ 2.29% | ↓ 6.36% |
| (c) | | ✔ | 61.92 | 55.92 | 48.10 | 60.66 | 60.40 | 57.40 |
| | | | ↓ 9.23% | ↓ 5.84% | ↓ 7.30% | ↓ 1.99% | ↓ 3.76% | ↓ 5.64% |
| (d) | ✔ | ✔ | **68.22** | **59.39** | **51.89** | **61.89** | **62.76** | **60.83** |

"Let's think step by step" to guide the model; and *Ours* refers to a complete multimodal thought-chain reasoning method enhanced by active learning. Compared with the baseline method, both the standalone Unc-Eva module and Com-Eva module enhance model accuracy in complex reasoning tasks to varying degrees. The Unc-Eva module demonstrates better performance across different disciplinary types and difficulty levels compared to the Com-Eva module. The Com-Eva module, while eliminating randomness, still improves model performance and ensures data stability and reliability. The experimental results indicate that the proposed method effectively combines the advantages of both modules, further enhancing model accuracy while mitigating randomness.

### 4.3.4 FOR Q4: CAMS CAN IMPROVE THE MODEL'S ACCURACY IN SUBDIVIDED DOMAINS.

To further investigate the efficacy of CAMS in specific subfields, we conduct systematic experiments on the ScienceQA dataset. Specifically, based on disciplinary attributes, the ScienceQA dataset is categorized into three major groups: natural sciences (NAT), social sciences (SOC), and linguistic sciences (LAN). According to difficulty gradients, it is also divided into two levels: grades 1–6 (G1-6) and grades 7–12 (G7-12), thus constructing multidimensional subscenarios that capture both domain specificity and cognitive complexity. As shown in Figure 4, in the five subfields mentioned above, CAMS almost universally outperforms the three baseline methods - Self-con, ZS-CoT, and FS-CoT - in terms of precision, especially in the NAT, SOC, and G1-6 domains. Whether in knowledge-centric scenarios emphasizing disciplinary dimensions or in learning settings focused on difficulty levels, CAMS demonstrates significant advantages, confirming its ability to substantially improve model accuracy within specialized domains. Meanwhile, this also demonstrates the generalization capability of the CAMS framework.

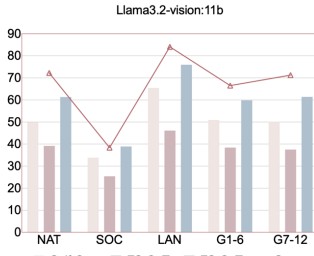 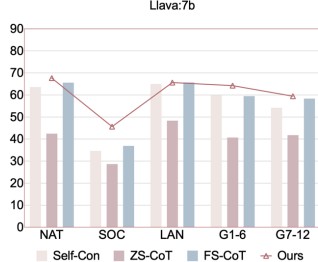 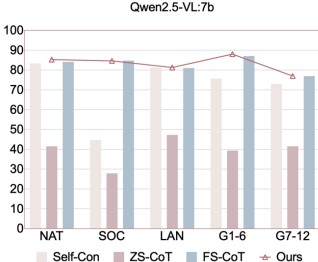

Figure 4: Comparison of accuracy between CAMS and three baseline methods in subdivided domains.

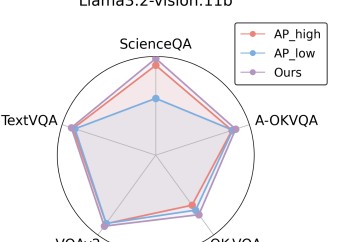 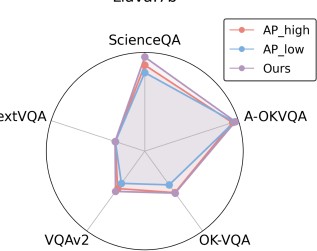 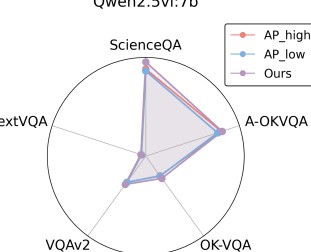

Figure 5: Comparison of different example selection strategies. "AP_high" denotes the selection of only difficult examples (i.e., those with high uncertainty); "AP_low" denotes the selection of only easy examples (i.e., those with low uncertainty).

### 4.3.5 FOR Q5: A SELECTION STRATEGY THAT COMBINES EASY AND DIFFICULT EXAMPLES IS MORE BENEFICIAL TO THE MODEL.

To evaluate the effectiveness of combining easy and difficult examples, we conducte additional experiments using two additional selection strategies: one that includes only difficult examples (i.e., those with high uncertainty) and another that includes only easy examples (i.e., those with low uncertainty). As shown in Figure 5, while the all-difficult example strategy performs relatively well across the five target datasets, it still falls short compared to the balanced strategy employed by CAMS. This finding highlights that relying exclusively on either high- or low-difficulty samples is insufficient to capture the full spectrum of knowledge complexity and scenario diversity required for complex reasoning tasks. In contrast, CAMS's approach of integrating both easy and difficult examples enables the model to learn more diverse and representative knowledge patterns, ultimately leading to improved reasoning accuracy.

## 5 CONCLUSION AND FUTURE WORK

In this work, we address key challenges in existing Chain-of-Thought (CoT) prompting methods for multimodal models, which often suffer from unstable and suboptimal performance due to their reliance on random or manually selected examples. To overcome these limitations, we propose CAMS, a novel framework inspired by the pedagogical principle of "tailored teaching with balanced difficulty". CAMS integrates two key dimensions — model-perceived difficulty and sample complexity — to construct a customized prompt curriculum that balances between easy and challenging examples. We demonstrate the effectiveness of CAMS through experiments on five benchmarks using multiple state-of-theart multimodal large language models. We also highlight promising directions for future work, including adapting CAMS to broader multimodal settings and further exploring dataset feature dimensions for deeper curriculum design.

REPRODUCIBILITY STATEMENT

We have submitted the relevant code in the supplementary materials. The names of the experimental benchmarks, the prompt templates used, and the model's hyperparameter settings can all be found in The Appendix A.1, A.2, A.3 and Table 7. Section 4 (Experiments) provides a detailed description of the experimental setup for the mechanism experiments.

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

## A  APPENDIX

### A.1  THE CONSTRUCTION OF PROMPT

We construct prompt examples by integrating questions, reasoning steps, and answers from the dataset:

$$E = \{(q_1, c_1, a_1), (q_2, c_2, a_2), ..., (q_n, c_n, a_n)\} \tag{7}$$

where $q$ denotes questions, $c$ denotes reasoning steps, $a$ denotes answers, and $n$ denotes the number of examples.

### A.2  COMPARISON BETWEEN CAMS AND AUTO-COT

Figure 6 demonstrates that the prompt examples selected by CAMS can, to a certain extent, enhance the strengths and mitigate the weaknesses of multimodal large models, thereby better facilitating their completion of reasoning tasks.

### A.3  PROMPT DEMONSTRATION

Details are provided in Table 7. It is important to note that the captions in the examples are replaced with actual images in real-world applications; text is used here solely for the convenience of demonstration.

Figure 6: Comparative cases between CAMS and Auto-CoT

## A.4 COMPLEXITY SCORER

Table 3 and Table 4 shows prompt templates for adding constraints and specification. Specification refers to increasing the depth and breadth of the problem, such as moving from a surface description to exploring its essence, or from a "single task/scenario" to a "multi-task/generalized scenario". Table 5 and Table 6 shows two examples of sample evolution.

## A.5 THE USE OF LARGE LANGUAGE MODELS (LLM)

In order to enhance the language quality and clarity of this academic paper, the author utilized AI-powered tools for text refinement during the writing process. The specific details are as follows:

**Purpose of Use:** The primary purposes for using AI tools were to:

Table 3: Prompt template of add constraints.

**SYSTEM:**
Prompt Templates of add constraints

**USER:**
I need you to play the role of a prompt rewriter.
Your task is to rewrite a given prompt into a more complex version that makes it more difficult for famous large language models like GPT4 to solve these problems.
The rewritten prompt must be reasonable, understandable and responded by humans.
Your rewrite cannot omit non-text parts, such as tables and code in #Given Prompt #.
Meanwhile, do not omit the input in #Given Prompt#.
You should use the following methods to complicate the given prompts:
Please add at least one restriction/requirement into #Given Prompt#.
You should try your best to avoid making #Rewritten Prompt# verbose.
#Rewritten Prompt# can only add 10 to 20 words into #Given Prompt#.
"#Rewritten Prompt#", "#Given Prompt#", "rewritten prompt" and "given prompt"' are not allowed to appear in #Rewritten Prompt#.
#Given Prompt#:
¡Here is instruction
#Rewritten Prompt#:

Table 4: Prompt template of specification

**SYSTEM:**
Prompt Templates of add constraints

**USER:**
I need you to play the role of a prompt rewriter.
Your task is to rewrite a given prompt into a more complex version that makes it more difficult for famous large language models like GPT4 to solve these problems.
The rewritten prompt must be reasonable, understandable and responded by humans.
Your rewrite cannot omit non-text parts, such as tables and code in #Given Prompt #.
Meanwhile, do not omit the input in #Given Prompt#.
You should use the following methods to complicate the given prompts:
If #Given Prompt# contains inquiries about certain issues, the depth and breadth of the inquiry can be increased.
You should try your best to avoid making #Rewritten Prompt# verbose.
#Rewritten Prompt# can only add 10 to 20 words into #Given Prompt#.
"#Rewritten Prompt#", "#Given Prompt#", "rewritten prompt" and "given prompt"' are not allowed to appear in #Rewritten Prompt#.
#Given Prompt#:
¡Here is instruction
#Rewritten Prompt#:

- Check grammar and spelling for certain sentences.

- Optimize vocabulary choices for more precise and academic expression.

- Adjust sentence structures to improve logical coherence and readability between paragraphs.

**Method of Use:** The author input original paragraphs written by themselves into the AI tools and then manually judged, filtered, and revised the text based on the refinement suggestions provided.

Table 5: Example of adding constraints.

| Instruction |
| --- |
| Find the title of a book about a political movement in the 1940s. |

| Response |
| --- |
| Find the title of a book about a 1940s anti-colonial political movement with first-hand activist testimonies. |
| Find the title of a 1940s anti-colonial political movement book set in Southeast Asia with first-hand activist testimonies and a major international literary award. |
| Find the title of a 1940s anti-colonial political movement book set in Southeast Asia, written by a local female activist (1945-1949), with first-hand activist testimonies and a major international literary award. |
| Find the title of a 1940s anti-colonial political movement book set in Southeast Asia, written by a local female activist (1945-1949), centered on rural grassroots mobilization and translated into three European languages, with first-hand activist testimonies and a major international literary award. |

Table 6: Example of specification.

| Instruction |
| --- |
| Find the title of a book about a political movement in the 1940s. |

| Response |
| --- |
| Find the title of a non-fiction book about a Southeast Asian political movement in the 1940s that shaped post-WWII governance. |
| Find the title of a non-fiction book about a Southeast Asian anti-colonial political movement in the 1940s that shaped post-WWII governance drawing on first-hand activist accounts. |
| Find the title of a non-fiction book about a Southeast Asian anti-colonial political movement in the 1940s that shaped post-WWII governance drawing on first-hand activist accounts, focused on Indochina or Indonesia and analyzed through transnational lenses. |
| Find the title of a non-fiction book about a Southeast Asian anti-colonial political movement in the 1940s that shaped post-WWII governance drawing on first-hand activist accounts, focused on Indochina or Indonesia and analyzed through transnational lenses, emphasizing marginalized women activists' roles in diplomatic shifts. |

All adopted changes were carefully considered by the author to ensure they fully align with the original intent and academic rigor of the paper.

**Disclaimer of Responsibility:** All academic content in this paper, including core arguments, research data, result analysis, argumentation process, and final conclusions, was independently created and is the sole responsibility of the author. The AI tools were used purely as an auxiliary aid and did not generate any critical academic viewpoints, research data, or conclusions. The author assumes full responsibility for the final content of the paper.

**Tools Used:** The AI tool used in this process is: GPT-4.

**Algorithm 1** CAMS

---

**Require:** Training set $D = \{x_1, x_2, ..., x_N\}$, Test set $T$, Multi-modal model $M$, Complexity scorer $C$, Number of iterations $k$, Sample size $n$
**Ensure:** Final accuracy on test set $T$
1: Initialize response log $R = \{\emptyset \mid x_i \in D\}$ {Empty list for each sample}
2: **for** $i = 1$ to $k$ **do**
3:    **for** each $x \in D$ **do**
4:       $y = M(x)$ {Get model's response to sample $x$}
5:       Append $y$ to $R[x]$ {Store response for sample $x$}
6:    **end for**
7: **end for**
8: Initialize disagreement scores $Disagreement = \{0 \mid x_i \in D\}$
9: **for** each $x \in D$ **do**
10:    $UniqueResponses = $ Remove duplicates from $R[x]$
11:    $k' = $ Length of $UniqueResponses$
12:    $Disagreement[x] = k'/k$
13: **end for**
14: $HardSamples = \{x \in D \mid 0.5 < Disagreement[x] \leq 1\}$
15: $EasySamples = \{x \in D \mid 0 < Disagreement[x] \leq 0.5\}$
16: **for** each $x \in HardSamples \cup EasySamples$ **do**
17:    $Complexity[x] = C(x)$ {Get complexity score from scorer}
18: **end for**
19: Sort $HardSamples$ by $Complexity[x]$ in ascending order
20: $HardLow = $ First $\frac{n}{4}$ samples from sorted $HardSamples$ {Low complexity hard samples}
21: $HardHigh = $ Last $\frac{n}{4}$ samples from sorted $HardSamples$ {High complexity hard samples}
22: Sort $EasySamples$ by $Complexity[x]$ in ascending order
23: $EasyLow = $ First $\frac{n}{4}$ samples from sorted $EasySamples$ {Low complexity easy samples}
24: $EasyHigh = $ Last $\frac{n}{4}$ samples from sorted $EasySamples$ {High complexity easy samples}
25: $Exemplars = HardLow \cup HardHigh \cup EasyLow \cup EasyHigh$
26: $Correct = 0$
27: **for** each $t \in T$ **do**
28:    $y_{pred} = M(t \mid Exemplars)$ {Model prediction with exemplars}
29:    **if** $y_{pred} == $ true label of $t$ **then**
30:       $Correct = Correct + 1$
31:    **end if**
32: **end for**
33: $Accuracy = Correct/$Length of $T$
34: **return** $Accuracy$

---

Table 7: Prompt template of query formulation.

**SYSTEM:**
You are a professional VQA task solver

**USER:**
Given the problem and its related images, you need to generate answers for the VQA task.

The generated answers must be derived from direct visual observation of the image—do not include speculative content, assumptions, or information not visible in the image.

When answering questions: if the image clearly shows the required information, output a specific, concise answer; if the image does not contain enough information to answer (e.g., object not visible, detail unclear), output "insufficient information"; if the image contradicts the question (e.g., question asks about a "red car" but image shows a blue car), output a negative answer with correct visual details.

Do not answer subjective judgment questions (e.g., "Is the image interesting?")—visual facts only.

You MUST only respond in the format as described below.

DO NOT RESPOND WITH ANYTHING ELSE.

Response Format:The answer is ..., because ... .

Here are an example:

"question":"Which of these organisms contains matter that was once part of the phytoplankton?"

"choices":["black rockfish", "sea otter"]

"hint":"Below is a food web from an ocean ecosystem in Monterey Bay, off the coast of California. A food web models how the matter eaten by organisms moves through an ecosystem. The arrows in a food web represent how matter moves between organisms in an ecosystem."

"caption":"A painting of a penguin on a wall."

"lecture":"A food web is a model. A food web shows where organisms in an ecosystem get their food. Models can make things in nature easier to understand because models can represent complex things in a simpler way. If a food web showed every organism in an ecosystem, the food web would be hard to understand. So, each food web shows how some organisms in an ecosystem can get their food. Arrows show how matter moves. A food web has arrows that point from one organism to another. Each arrow shows the direction that matter moves when one organism eats another organism. An arrow starts from the organism that is eaten. The arrow points to the organism that is doing the eating. An organism in a food web can have more than one arrow pointing from it. This shows that the organism is eaten by more than one other organism in the food web. An organism in a food web can also have more than one arrow pointing to it. This shows that the organism eats more than one other organism in the food web."

"output":"The answer is A, because Use the arrows to follow how matter moves through this food web. For each answer choice, try to find a path of arrows that starts from the phytoplankton. The only arrow pointing to the sea otter starts from the sea urchin. The only arrow pointing to the sea urchin starts from the kelp. No arrow points to the kelp. So, in this food web, matter does not move from the phytoplankton to the sea otter.There are two paths matter can take from the phytoplankton to the plainfin midshipman: phytoplankton− >plainfin midshipman. phytoplankton− >zooplankton− >plainfin midshipman. There is one path matter can take from the phytoplankton to the black rockfish: phytoplankton− >zooplankton− >black rockfish. There is one path matter can take from the phytoplankton to the zooplankton: phytoplankton− >zooplankton."

Now complete your output with following the above rules.

Input:

Output:

