# OpenReview forum: "Tailored Teaching with Balanced Difficulty: Elevating Reasoning in Multimodal Chain-of-Thought via Prompt Curriculum"
_ICLR.cc/2026/Conference — ICLR 2026 Conference Withdrawn Submission_

### Official Review · Reviewer_CvGg · 2025-10-17

**Soundness:** 3
**Presentation:** 2
**Contribution:** 3
**Rating:** 6
**Confidence:** 4

**Summary:**

This paper introduces a novel framework named CAMS to address the performance instability of Multimodal Chain-of-Thought (MCOT) caused by its reliance on randomly selected examples. Inspired by the principle of "tailored teaching," CAMS reframes prompt selection as a systematic "prompt curriculum design" problem. It innovatively selects examples by considering two dimensions: 1) model-perceived difficulty (uncertainty), measured by the model's own prediction disagreement, and 2) intrinsic sample difficulty (complexity), judged by an independent assessor. By balancing examples from these two dimensions, the method demonstrates significant and stable performance improvements in reasoning across multiple benchmarks.

**Strengths:**

1. The work's primary novelty lies in creatively fusing two distinct concepts—the model-centric view of uncertainty and the data-centric view of complexity—into a single, unified framework for example selection. Elevating prompt selection to "prompt curriculum design" provides a fresh and systematic perspective for the field.

2. The experimental design is  robust. Comprehensive evaluations across five diverse benchmarks and three different mainstream models strongly support the method's generalization ability and model-agnosticism. The thorough ablation studies also clearly validate the necessity of each core component of the framework.

3. The paper is well-structured and highly readable.

4. This research directly addresses a critical pain point in current large model applications: prompt fragility and instability. It provides a systematic, reproducible solution that significantly enhances the reliability and performance of MCOT for complex reasoning, holding substantial practical value.

**Weaknesses:**

1. Limited Scope of the Complexity Assessor and Its Potential Impact on Generalization: The paper's core innovation, the "complexity" dimension, is defined in a way that equates complexity with structural difficulty (i.e., more reasoning steps). This approach is effective for problems where complexity and difficulty are aligned. However, it may fail on tasks where difficulty stems from conceptual insight rather than procedural length. The paper's own results (Figure 4) hint at this, showing a smaller performance gain in the language science (LAN) domain. This raises concerns about the method's effectiveness on higher-level reasoning tasks where a key insight, not structural complexity, is the primary barrier.

2. Lack of Transparency and Reproducibility in the Complexity Assessor's Training: A key step is the "Evol-Complexity" method, which uses ChatGPT to iteratively increase question difficulty to generate training data. However, the paper does not provide the specific prompts and examples used to guide this process. This makes the core module a "black box," hindering reproducibility and preventing a deeper analysis of the biases the assessor might have learned from ChatGPT.

3. High Computational Cost is Undiscussed: The CAMS framework requires a computationally intensive preprocessing stage, involving multiple inference passes for uncertainty estimation and calls to another large model for complexity assessment. The paper focuses exclusively on performance gains without providing any discussion on the trade-off between these gains and the significant increase in computational overhead, which is a critical factor for practical application.

**Questions:**

1. On the Effective Boundary of the Complexity Assessor: We observed in Figure 4 that the performance gain in the language science (LAN) domain was less pronounced than in NAT and SOC. Could this suggest that your complexity assessor, trained by increasing "reasoning steps," is less effective at capturing the "conceptual" or "insight-based" difficulty common in language problems? Following this, how would you anticipate your method performing on high-level reasoning tasks like AIME math problems, where difficulty is almost entirely conceptual? Have you considered alternative or complementary methods to assess complexity for such challenges?

2. On the Reproducibility of "Evol-Complexity": To improve the transparency and reproducibility of your work, would it be possible to provide the specific prompt used to guide ChatGPT in increasing question complexity, along with one or two full examples showing a question's "evolution" through the iterative process?

3. On the Consideration of Computational Cost: Could you provide a brief quantitative analysis of the computational overhead (e.g., approximate extra GPU hours or API calls) required by the CAMS preprocessing pipeline compared to a baseline like FS-CoT? This information is crucial for readers to assess the practical deployment value of your method.

---

> ### Author Response · Authors · 2025-11-19
> **Response to Reviewer CvGg**
>
> Dear reviewer CvGg, thank you for your patient review. We are honored to answer your questions.
>
> ### For Weakness 1
>
> We want to clarify that **we are using the full CAMS here, and as shown in Figure 4, our method outperforms the baseline model in all domains.** As you mentioned, the improvement of CAMS in the "LAN" domain is not as significant as in other domains. **We will focus on investigating the factors that cause this in later experiments and explore how to improve it.**
>
> ### For Weakness 2
>
> In response to your request, **we have provided detailed templates for sample evolution and scoring, as well as examples of adding constraints and specification, in "Response to all reviewers about common questions (Part 2/3): A4" and "Response to all reviewers about common questions (Part 3/3): A5"**. We hope you find this helpful.
>
> ### For Weakness 3
>
> We describe the computational cost in detail in "Response to all reviewers about common questions (Part 2/3): A3". Furthermore, when dealing with datasets like VQAv2 with hundreds of thousands of training samples, as shown in Table 1, CAMS achieves significant results even with our 10k upper limit. **Therefore, the computational cost of CAMS is controllable.**

---

> > ### Comment · Reviewer_CvGg · 2025-11-24
> >
> > Thank you for your reply. It has addressed some of my questions, and I've decided to keep the score unchanged.

---

### Official Review · Reviewer_UdSN · 2025-10-29

**Soundness:** 3
**Presentation:** 3
**Contribution:** 2
**Rating:** 6
**Confidence:** 3

**Summary:**

The paper proposes CAMS (Complexity-Guided Active Multimodal CoT Sampling), a prompt–curriculum framework for multimodal chain-of-thought (MCoT) reasoning. Instead of randomly or manually picking few-shot exemplars, CAMS selects a balanced set of examples by combining: (i) model-perceived difficulty via prediction disagreement across k stochastic forward passes (unique answers divided by k), and (ii) intrinsic sample complexity estimated by a learned complexity scorer (trained on “evolved” instruction variants and applied to text formed by concatenating question, options, and image caption). Samples are partitioned into easy/hard by the disagreement score and into low/high complexity by the scorer; CAMS then samples equally from the four buckets to build the prompt exemplars (shared across all test items). Experiments on ScienceQA, A-OKVQA, OK-VQA, VQAv2, TextVQA with LLaMA3.2-Vision-11B, LLaVA-7B, Qwen2.5-VL-7B report consistent gains over ZS-CoT, FS-CoT, Auto-CoT, Active Prompt, and Self-Consistency, plus reduced variance vs random few-shot selection. Ablations indicate both uncertainty and complexity contribute, and analyses on ScienceQA sub-domains suggest broad benefits.

**Strengths:**

1. Clear framing of prompt selection as a tailored curriculum with two orthogonal signals (model-uncertainty & data-complexity) and a balanced difficulty principle; simple and general recipe usable across MLLMs.

2. Multi-benchmark, multi-model evaluation; ablations showing both modules matter; analysis on sub-domains (ScienceQA NAT/SOC/LAN; grade bands). Reported average gains over strong baselines plus reduced seed-variance.

3. Method flow (Fig. 2) and pseudocode (Algorithm 1) make the approach reproducible; implementation details (temperature, k, sampling limits) are provided; prompt templates are in the appendix; code promised.

4. Addresses a practical pain point—random exemplars cause instability—and offers a principled fix with measurable benefits for MCoT on common VQA-style tasks.

**Weaknesses:**

1. The scorer depends on caption quality and a text-only transformation of multimodal inputs; evidence that this approximates intrinsic multimodal complexity is limited. A calibration study (e.g., correlation with human difficulty ratings or model error rates) would strengthen the claim.

2. The 0.5 uncertainty threshold, equal bucket quotas, and exemplar count are not stress-tested. Without sensitivity curves, it’s hard to disentangle whether “balanced difficulty” per se or just “not all-hard” drives gains.

3. Active disagreement over large pools (even with 10k caps) can be expensive; there is no cost vs. accuracy analysis or amortization strategy (e.g., sub-sampling, early stopping, proxy models).

4. While Auto-CoT, Self-Consistency, and Active Prompt are included, comparisons to complementary planning/verification paradigms (e.g., Least-to-Most, ReAct/ToT in multimodal settings) are absent; also no results with larger closed models to assess headroom.

5. Needs explicit confirmation that official VQA scoring (soft-acc, text normalization) is used; if not, comparisons may be apples-to-oranges.

6. The construction of exemplars by concatenating dataset “question, reasoning, answer” presumes such rationales exist and are high-quality for each dataset; many VQA datasets lack gold rationales. Scope and handling when rationales are unavailable should be clarified.

**Questions:**

See above

---

> ### Author Response · Authors · 2025-11-19
> **Response to Reviewer UdSN**
>
> Dear reviewer UdSN, thank you for your patient review. We are honored to answer your questions.
>
> ### For Weakness 1
>
> This is a very worthwhile question to study. **We are using image captions created by benchmark creators, which accurately describe the images and reflect their complexity.** Unfortunately, due to insufficient manpower, we are currently unable to conduct the calibration study you proposed; we will address this later.
>
> ### For Weakness 2
>
> We conducted extensive supplementary experiments. The threshold is shown in "Response to all reviewers about common questions (Part 1/3): A1", and the mixing ratio in "Response to all reviewers about common questions (Part 1/3): A2". **The experiments demonstrate that the threshold and mixing ratio we selected are optimal.**
>
> ### For Weakness 3
>
> We quantified the computational cost in "Response to all reviewers about common questions (Part 2/3): A3", and the experiments in Table 1 of the paper also verify that when dealing with the VQAv2 dataset with hundreds of thousands of training samples, randomly selecting 10,000 samples for uncertainty analysis still yields significant and stable results. **In conclusion, the computational cost of CAMS is controllable.**
>
> ### For Weakness 4
>
> Auto-Cot, self-consistency, and Active Prompt **are all classic prompting word methods**, which is why we chose these methods as our baseline. Meanwhile, the Llama, Lava, and Qwen models we selected **are all classic and well-known multimodal models.** As for larger closed-source models, due to funding constraints, we are currently unable to deploy them for experiments.
>
> ### For Weakness 5
>
> **The ScienceQA benchmark uses accuracy as the metric in its test code template**, so we continued to use this metric in our subsequent experiments.
>
> ### For Weakness 6
>
> **Our research focuses on the selection of prompts** rather than the writing of prompts. In our experiments, OK-VQA, VQAv2, and TextVQA did not use reasoning; their prompts were simply a combination of questions and answers. However, **as shown in Table 1 of the paper, CAMS also achieved significant results.**

---

### Official Review · Reviewer_U7iv · 2025-10-30

**Soundness:** 2
**Presentation:** 2
**Contribution:** 2
**Rating:** 2
**Confidence:** 3

**Summary:**

This paper addresses the suboptimal and unstable performance of Multimodal Chain-of-Thought (MCoT) prompting, which it attributes to the reliance on randomly or manually selected examples. The authors propose CAMS, a framework to automate prompt selection by constructing a "prompt curriculum" based on two signals: model-perceived difficulty, measured via prediction uncertainty, and intrinsic sample complexity, determined by a separate scorer. By jointly analyzing these dimensions, the method selects a balanced set of examples intended to be more effective. The approach is evaluated on five multimodal reasoning benchmarks, with the authors reporting improved accuracy and reduced performance variance compared to baseline prompting strategies.

**Strengths:**

- The paper uniquely reframes MCoT prompt selection as a curriculum design problem. It innovatively combines two complementary signals—model-specific uncertainty and model-agnostic sample complexity—to create a more principled and effective selection strategy.

- The empirical validation is robust, featuring extensive experiments on five benchmarks with three different MLLMs. The ablation studies are a key strength, clearly demonstrating the synergistic benefit of the two signals, and the work provides strong quantitative evidence of improved performance stability.

**Weaknesses:**

- A core component of the proposed framework is the "Complexity Scorer," yet its development and impact are not fully interrogated. The scorer is trained on a dataset created via an "evolution-based metric" that relies on ChatGPT for ranking. This introduces several issues: (1) It creates a dependency on a powerful, proprietary model, which complicates reproducibility. (2) The performance of CAMS becomes implicitly dependent on the quality of this external LLM's judgments. The paper lacks a sensitivity analysis exploring how the final MCoT performance is affected by the accuracy of this scorer.

- The paper's key claim is that a "balanced difficulty" curriculum is superior. However, the implementation of this balance—selecting an equal number of samples from four discrete quadrants (easy/hard uncertainty vs. easy/hard complexity)—is a rigid heuristic. It is not obvious that a 25/25/25/25 split is optimal across different models, which have unique knowledge gaps, or across different datasets with varying intrinsic difficulty distributions.

**Questions:**

Plz answer my concerns in the weakness section

---

> ### Author Response · Authors · 2025-11-19
> **Response to Reviewer U7iv**
>
> Dear reviewer U7iv, thank you for your patient review. We are honored to answer your questions.
>
> ### For Weakness 1
>
> In "Response to all reviewers about common questions (Part 2/3): A4" and "Response to all reviewers about common questions (Part 3/3): A5", we have listed the prompt templates for sample evolution and scoring, as well as sample demonstrations for adding constraints and specification. Using these templates, **anyone can create a unique dataset of their own sample evolutions. Based on this dataset, any model can be trained as a complexity scorer, thus eliminating any difficulty in reproduction.** In Section 3.3, we mentioned that this complexity scorer achieved an **AlpacaEva score of 90.06%, indicating extremely high scoring quality.** Furthermore, as shown in Table 2, in our ablation experiments, even without this crucial complexity scoring step, the model's accuracy still showed a significant improvement. **Therefore, even without this complexity scorer, CAMS still has a significant effect, not to mention that this complexity scorer has excellent performance.**
>
> ### For Weakness 2
>
> To address this, we conducted a supplementary experiment, as shown in "Response to all reviewers about common questions (Part 1/3): A2". For the proportions in each of the four quadrants, we listed all 35 different mixing ratios and experimented with them all. **This experiment demonstrates that a mixing ratio of 1:1:1:1 is optimal.**

---

> ### Author Response · Authors · 2025-11-28
> **Gentle reminder regarding our response**
>
> We truly appreciate the effort you put into reviewing our paper.
>
> We wanted to gently check if you have seen our detailed response to your comments. We have made significant efforts to clarify the contribution of CAMS and provided the quantitative analysis on computational costs and hyperparameters that you requested.
>
> We hope our response helps clarify the misunderstandings, and we look forward to hearing your thoughts.
>
> Best regards, The Authors

---

### Official Review · Reviewer_eDCG · 2025-11-05

**Soundness:** 1
**Presentation:** 2
**Contribution:** 1
**Rating:** 2
**Confidence:** 4

**Summary:**

This paper proposes CAMS, a prompt demonstration selection framework for multimodal CoT reasoning. CAMS uses a disagreement-based uncertainty score and a complexity scorer to build the example pool. Experiments on several VQA benchmarks show gains over baseline CoT methods. However, key design choices are ad hoc, and the complexity signal is only weakly justified, and there is no serious analysis. The contribution mainly refines existing ideas in LLMs.

**Strengths:**

1. This paper targets a practical issue that how to construct the prompt demonstration for multimodal CoT reasoning.
2. The experiments cover multiple VQA benchmarks and several multimodal models, showing the effectiveness of  the proposed CAMS framework.
3. The overall pipeline is relatively easy to follow and easy to implement.

**Weaknesses:**

1. The core contribution feels like a combination of existing ideas in LLMs rather than a new concept for multimodality. The improvement over simple baselines is often modest. The proposed method also introduces additional computation cost in prompt selection.
2. The proposed complexity scorer is trained on text-only data using an outdated LLM (llama-2) and applied to caption-plus-question inputs for multimodal tasks, but this paper does not offer convincing evidence that this score meaningfully reflects sample difficulty for these VQA benchmarks.
3. Key choices such as the disagreement threshold for splitting easy and hard examples and the fixed mixing of high- and low-complexity samples are not thoroughly studied. There is no sensitivity analysis that shows the method is robust to these hyperparameters.
4. It would be better to see quantitative discussion of computational cost because CAMS requires multiple passes of a large MLLM over a training subset and additional passes of a complexity scorer.  For realistic scenario, processing the whole training dataset may be costly or infeasible.
5. I would like to know why CAMS constructs a single global prompt pool per dataset and does not adapt exemplars to individual test questions, which contradicts the "tailored".

**Questions:**

Please see the weakness.

---

> ### Author Response · Authors · 2025-11-19
> **Response to Reviewer eDCG**
>
> Dear reviewer eDCG, thank you for your patient review. We are honored to answer your questions.
>
> ### For Weakness 1
>
> **We apologize for any misunderstanding caused by our misstatement. Our method is absolutely not an integration of existing methods.** **As illustrated in the Section 1 and Figure 1**, classic prompting methods, whether Active Prompt, Auto-CoT, or few-shot prompts using randomly selected or manually written examples, either focus solely on the knowledge distribution within the model, the characteristics of the dataset itself, or neither. **Our proposed CAMS is a novel prompting method that considers both the knowledge distribution within the model and the characteristics of the dataset itself**. We do not consider this an integration of existing methods, but rather **a completely new approach addressing the pain points of existing multimodal large-scale models and VQA benchmarks**. Regarding the improvement results, As shown in Table 1, the quantitative results indicate that CAMS's improvement over simple baselines is not "limited," but rather **exhibits a significant and stable improvement in multimodal complex inference scenarios.** More importantly, as shown in Figure 3, **CAMS's improvement is not only reflected in "accuracy values," but also addresses the core deficiency of simple baselines in multimodal scenarios—lack of stability.**
>
> ### For Weakness 2
>
> We trained llama2 as a complexity scorer on a created dataset. **Llama2 is a classic and excellent large language model, therefore it does not have the problem of being outdated.** It's worth mentioning that, as we explained in Section 3.3 of the paper, when we scored the complexity of the training samples on the VQA benchmarks, **we used image captions and corresponding question and answer texts created by the benchmark's creators.** This content is of high quality and accurate. **Also in Section 3.3,** **we mentioned that this complexity scorer achieved an AlpacaEva score of 90.06%**. Therefore, this complexity scorer can accurately reflect the complexity of these VQA samples.
>
> ### For Weakness 3
>
> We have conducted supplementary experiments; details of the thresholds and the proportions of each type of sample can be found in "Response to all reviewers about common questions (Part 1/3)". **The experiments demonstrate that the threshold and mixing ratio we selected are optimal.**
>
> ### For Weakness 4
>
> The analysis of computational costs can be found in "Response to all reviewers about common questions (Part 2/3): A3"; **we have proven that computational costs are controllable.** Meanwhile, the additional complexity scorer is reusable; only one needs to be trained to be applied to the rest of the dataset.
>
> ### For Weakness 5
>
> Applying individual prompts to each test sample requires massive computational costs. Imagine the test set as an exam, the multimodal large language model as a test-taker, and the training set as all the practice questions for that exam. Each test-taker has different knowledge bases, strengths, and weaknesses, and therefore requires different prompts. **For this exam and for each individual test-taker, we provide different prompts—this is precisely a kind of "tailored" approach.**

---

> ### Author Response · Authors · 2025-11-28
> **Gentle reminder regarding our response**
>
> Dear Reviewer eDCG,
>
> We truly appreciate the effort you put into reviewing our paper.
>
> We wanted to gently check if you have seen our detailed response to your comments. We have made significant efforts to clarify the contribution of CAMS and provided the quantitative analysis on computational costs and hyperparameters that you requested.
>
> We hope our response helps clarify the misunderstandings, and we look forward to hearing your thoughts.
>
> Best regards,
> The Authors

---

### Author Response · Authors · 2025-11-19
**Response to all reviewers about common questions (Part 1/3)**

Dear reviewers, thank you for your patient review. We are honored to answer your questions. The following prompt templates and examples of sample evolution for the complexity scorer  have been **updated in Appendix A4 and Tables 3-6 of the revised version.**

First, we will answer some common questions.

# A. Common questions

## A1. Threshold for classifying difficult and easy questions

| Threshold | 0.2   | 0.4   | 0.5       | 0.6   | 0.8  |
| --------- | ----- | ----- | --------- | ----- | ---- |
| Accuracy  | 47.19 | 45.54 | **51.89** | 48.87 | 49.8 |

Thank you very much for your question about the threshold. We sincerely accepted your suggestion and conducted experiments. As shown in the table, in addition to setting the threshold to 0.5, we also set it to 0.2, 0.4, 0.6, and 0.8. **It can be seen that setting the threshold to 0.5 yielded a significant advantage.** **Therefore, we can conclude that a threshold setting of 0.5 is an excellent choice.**

## A2. The ratio of difficult problems, easy problems, high-complexity problems, and low-complexity problems

We believe this is a very worthwhile question to study, so we conducted a large number of supplementary experiments. The specific experimental results are shown in the following tables. **It is important to note that since we have already verified in A1 that setting the threshold to 0.5 is an excellent choice, we maintained the threshold of 0.5 in this experiment.**

| High Complexity & Difficulty : Low Complexity & Difficulty : High Complexity &  Easy : High Complexity & Easy | Accuracy  |
| ------------------------------------------------------------ | --------- |
| 4 : 0 : 0 : 0                                                | 45.28     |
| 0 : 4 : 0 : 0                                                | 43.58     |
| 0 : 0 : 4 : 0                                                | 48.93     |
| 0 : 0 : 0 : 4                                                | 47.13     |
| 3 : 1 : 0 : 0                                                | 46.20     |
| 3 : 0 : 1 : 0                                                | 45.85     |
| 3 : 0 : 0 : 1                                                | 45.03     |
| 1 : 3 : 0 : 0                                                | 46.03     |
| 0 : 3 : 1 : 0                                                | 45.28     |
| 0 : 3 : 0 : 1                                                | 45.35     |
| 1 : 0 : 3 : 0                                                | 45.43     |
| 0 : 1 : 3 : 0                                                | 46.58     |
| 0 : 0 : 3 : 1                                                | 47.77     |
| 1 : 0 : 0 : 3                                                | 46.54     |
| 0 : 1 : 0 : 3                                                | 46.75     |
| 0 : 0 : 1 : 3                                                | 47.84     |
| 2 : 2 : 0 : 0                                                | 47.12     |
| 2 : 0 : 2 : 0                                                | 47.51     |
| 2 : 0 : 0 : 2                                                | 47.28     |
| 0 : 2 : 2 : 0                                                | 47.99     |
| 0 : 2 : 0 : 2                                                | 46.98     |
| 0 : 0 : 2 : 2                                                | 48.44     |
| 2 : 1 : 1 : 0                                                | 47.55     |
| 2 : 1 : 0 : 1                                                | 46.15     |
| 2 : 0 : 1 : 1                                                | 46.10     |
| 1 : 2 : 1 : 0                                                | 46.21     |
| 1 : 2 : 0 : 1                                                | 45.28     |
| 0 : 2 : 1 : 1                                                | 47.32     |
| 1 : 1 : 2 : 0                                                | 46.09     |
| 1 : 0 : 2 : 1                                                | 45.62     |
| 0 : 1 : 2 : 1                                                | 46.91     |
| 1 : 1 : 0 : 2                                                | 47.35     |
| 1 : 0 : 1 : 2                                                | 45.18     |
| 0 : 1 : 1 : 2                                                | 46.07     |
| 1 : 1 : 1 : 1                                                | **51.89** |

**We combined the proportions of four quadrants: high complexity & difficult problems, low complexity & difficult problems, high complexity & easy problems, and low complexity & easy problems, resulting in a total of 35 combinations.** We experimented with all 35 combinations, and as can be seen from the table, **the 1:1:1:1 combination we chose in our paper is significantly better than the other combinations.**

---

### Author Response · Authors · 2025-11-19
**Response to all reviewers about common questions (Part 2/3)**

## A3. Computational cost

Computational cost is always a metric worth considering.  **It is important to note that, unlike selecting prompts for each test sample [1], CAMS only samples from the training samples, regardless of the number of test samples.** Assuming we perform *k* samplings and have *m* training samples, even with complexity scoring, we only need to perform *(k+1)\*m* samplings. Furthermore, when dealing with datasets like VQAv2 with hundreds of thousands of training samples, we still achieved a certain level of accuracy improvement by randomly selecting 10,000 samples. **In conclusion, we believe the computational cost of CAMS is** **controllable.**

[1] Exploring the Role of Diversity in Example Selection for In-Context Learning, SIGIR 2025

## A4. Complexity scorer

### The prompt template of the sample evolution

**Prompt Templates of add constraints**

```
I need you to play the role of a prompt rewriter.
Your task is to rewrite a given prompt into a more complex version that makes it more difficult for famous large language models like GPT4 to solve these problems.
The rewritten prompt must be reasonable, understandable and responded by humans.
Your rewrite cannot omit non-text parts, such as tables and code in #Given Prompt#.
Meanwhile, do not omit the input in #Given Prompt#.
You should use the following methods to complicate the given prompts:
Please add at least one restriction/requirement into #Given Prompt#.
You should try your best to avoid making #Rewritten Prompt# verbose.
#Rewritten Prompt# can only add 10 to 20 words into #Given Prompt#.
'#Rewritten Prompt#', '#Given Prompt#', 'rewritten prompt' and 'given prompt' are not allowed to appear in #Rewritten Prompt#.
#Given Prompt#:
<Here is instruction\>
#Rewritten Prompt#:
```

**Prompt Templates of specification**

```
I need you to play the role of a prompt rewriter.
Your task is to rewrite a given prompt into a more complex version that makes it more difficult for famous large language models like GPT4 to solve these problems.
The rewritten prompt must be reasonable, understandable and responded by humans.
Your rewrite cannot omit non-text parts, such as tables and code in #Given Prompt#.
Meanwhile, do not omit the input in #Given Prompt#.
You should use the following methods to complicate the given prompts:
If #Given Prompt# contains inquiries about certain issues, the depth and breadth of the inquiry can be increased.
You should try your best to avoid making #Rewritten Prompt# verbose.
#Rewritten Prompt# can only add 10 to 20 words into #Given Prompt#.
'#Rewritten Prompt#', '#Given Prompt#', 'rewritten prompt' and 'given prompt' are not allowed to appear in #Rewritten Prompt#.
#Given Prompt#:
<Here is instruction\>
#Rewritten Prompt#:
```

Specification refers to increasing the depth and breadth of the problem, **such as moving from a surface description to exploring its essence, or from a "single task/scenario" to a "multi-task/generalized scenario".**


### The prompt template of scoring

```
Please assess and rate the difficulty and complexity of the following questions. Please give a total score on a scale of 1 to 10, with higher scores indicating greater difficulty and complexity. You only need to provide a score; no further explanation is required.
Question: <Instruction\>
Score:
```

---

### Author Response · Authors · 2025-11-19
**Response to all reviewers about common questions (Part 3/3)**

## A5.Example of sample evolution

Here we take "Find the title of a book about a political movement in the 1940s" as an example, and evolve it by adding constraints and concretization (such as from surface description to essential exploration, from a single scenario to multiple scenarios). The following are the results of five iterations. The iterative results of the demonstration examples have been **updated in Appendix A4 and Tables 5-6 of the revised version.**

### Example of adding constraints

Increasing sample complexity by adding constraints

| **Instruction**                                              |
| ------------------------------------------------------------ |
| Find the title of a book about a political movement in the 1940s. |
| **Response**                                                 |
| Find the title of a book about a 1940s anti-colonial political movement with first-hand activist testimonies. |
| Find the title of a 1940s anti-colonial political movement book set in Southeast Asia with first-hand activist testimonies and a major international literary award. |
| Find the title of a 1940s anti-colonial political movement book set in Southeast Asia, written by a local female activist (1945-1949), with first-hand activist testimonies and a major international literary award. |
| Find the title of a 1940s anti-colonial political movement book set in Southeast Asia, written by a local female activist (1945-1949), centered on rural grassroots mobilization and translated into three European languages, with first-hand activist testimonies and a major international literary award. |


### Example of specification


Increase sample complexity by making it more specific.

| **Instruction**                                              |
| ------------------------------------------------------------ |
| Find the title of a book about a political movement in the 1940s. |
| **Response**                                                 |
| Find the title of a non-fiction book about a Southeast Asian political movement in the 1940s that shaped post-WWII governance. |
| Find the title of a non-fiction book about a Southeast Asian anti-colonial political movement in the 1940s that shaped post-WWII governance drawing on first-hand activist accounts. |
| Find the title of a non-fiction book about a Southeast Asian anti-colonial political movement in the 1940s that shaped post-WWII governance drawing on first-hand activist accounts, focused on Indochina or Indonesia and analyzed through transnational lenses. |
| Find the title of a non-fiction book about a Southeast Asian anti-colonial political movement in the 1940s that shaped post-WWII governance drawing on first-hand activist accounts, focused on Indochina or Indonesia and analyzed through transnational lenses, emphasizing marginalized women activists’ roles in diplomatic shifts. |

###

---

### Author Response · Authors · 2025-11-23
**Looking forward to for All Reviewers' Feedback and Discussion**

Dear all Reviewers,

We sincerely appreciate the constructive comments and insightful suggestions you have provided for our work. As the deadline for the discussion period approaches, we kindly encourage your continued engagement in the discussion and would greatly value any additional insights or clarifications you may wish to share.

Should you have further questions or require any additional information, please do not hesitate to reach out to us. We are more than willing to address any concerns, as your expertise is invaluable to us. We are confident that your input will significantly contribute to improving the quality of our work.

If our responses have addressed some of your concerns, we would be truly grateful if you could consider revising your rating score for our paper. Your support and consideration mean a great deal to us.

Thank you once again for your time and thoughtful feedback. We look forward to hearing from you soon.

Best regards,

All authors

---

### Author Response · Authors · 2025-12-02
**Summary comment prepared for AC, SAC and PC**

Dear AC, SAC and PC,

Thank you sincerely for your time and effort, **especially under the unusual circumstances of the current ICLR cycle.** We greatly appreciate your work overseeing our submission and managing the substantial additional load during this period.

To help you quickly understand our previous discussions with the reviewers, we have listed **the following common issues**: (1) the threshold for distinguishing between difficult and easy samples; (2) the mixing ratio of the four dimensions of difficulty, ease, high complexity, and low complexity; (3) computational cost; and (4) the construction of evolutionary samples for the complexity scorer.

**Our solutions** to these common problems are as follows:

**(1) Threshold**, as shown in "Response to all reviewers about common questions (Part 1/3): A1", we additionally set the threshold to 0.2, 0.4, 0.6 and 0.8 and conducted experiments. **The experimental results show that the threshold of 0.5 we selected achieved the best results.**

**(2) Mixing ratio**, as shown in "Response to all reviewers about common questions (Part 1/3): A2", we listed all 35 combinations and conducted experiments. **The results showed that the 1:1:1:1 mixing ratio we selected achieved the best results.**

**(3) Computational cost**, as shown in "Response to all reviewers about common questions (Part 2/3): A3", we have demonstrated that **the computational cost of CAMS is ideal and controllable.**

**(4) Construction of evolutionary samples.** As shown in "Response to all reviewers about common questions (Part 2/3): A4", we show the evolutionary samples and the prompt templates used for sample scoring. In "Response to all reviewers about common questions (Part 3/3): A5", we take "Find the title of a book about a political movement in the 1940s" as an example to show in detail the process of the five fireworks in the sample.

Meanwhile, for your convenience, we provide below a concise summary of each reviewer's post-rebuttal stance:

• *Reviewer UdSN:* Although this reviewer did not participate in the discussion, **Reviewer UdSN** had already **provided a positive recommendation from the beginning.**(**score:6**)

• *Reviewer UdSN:* After our clarification, the reviewer stated that we had addressed most of the issues he raised and chose to **maintain the original score**.**(****score:6**)

*"Thank you for your reply. It has addressed some of my questions, and I've decided to keep the score unchanged."*

• *Reviewer eDCG:* **Despite our extensive experimental work to address the questions raised**, we have received no response.(score:2)

• *Reviewer U7iv:* **Despite our extensive experimental work to address the questions raised**, we have received no response.(score:2)

In summary, We have thoroughly **addressed the four key common issues** raised by reviewers **with comprehensive experiments and detailed clarifications**, which have been validated by positive outcomes and acknowledged by the reviewer who **maintained his positive scores**. While three reviewers have not yet responded, we have spared no effort to address their concerns through extensive experimental work.

Thank you again for your time and for guiding our submission through this unusual review cycle. We hope the above summary helps streamline your decision process.

Warm regards,

The Authors

---

### Note · Authors · 2026-01-06

I have read and agree with the venue's withdrawal policy on behalf of myself and my co-authors.